# Elastic strain-induced amorphization in high-entropy alloys

Yeqiang Bu [1,2,9], Yuan Wu [3,4,9] ✉, Zhifeng Lei [5], Xiaoyuan Yuan[3], Leqing Liu[3], Peng Wang[6], Xiongjun Liu [3], Honghui Wu [3], Jiabin Liu [1,7] ✉, Hongtao Wang [1,2] ✉, R. O. Ritchie [8], Zhaoping Lu [3] ✉ & Wei Yang[1]

Elastic stability is the basis for understanding structural responses to external stimuli in crystalline solids, including melting, incipient plasticity and fracture. In this work, elastic stability is investigated in a series of high-entropy alloys (HEAs) using in situ mechanical tests and atomic-resolution characterization in transmission electron microscopy. Under tensile loading, the HEA lattices are observed to undergo a sudden loss of ordering as the elastic strain reached ∽10%. Such elastic strain-induced amorphization stands in intrinsic contrast to previously reported dislocation-mediated elastic instability and defect accumulation-mediated amorphization, introducing a form of elastic instability. Together with the first principle calculations and atomic-resolution chemical mapping, we identify that the elastic strain-induced amorphization is closely related to the depressed dislocation nucleation due to the local atomic environment inhomogeneity of HEAs. Our findings provide insights for the understanding of the fundamental nature of physical mechanical phenomena like elastic instability and incipient plasticity.

The structural response of crystalline solids to external stimuli such as heat and load generally determines their physical and mechanical behaviors, including melting, incipient plasticity, and fracture[1–4]. High-entropy alloys (HEAs) contain multiple principal constituents and possess a scattering-distributed bond length due to their large mismatch of atomic sizes and chemical complexities[5,6]. Due to the severe heterogeneity of chemical bonds, HEAs are considered to have intrinsically different structural responses to the external stimuli as compared with traditional dilute alloys. However, studies have shown that the incipient plastic deformation mechanism of bulk HEAs is still mediated by dislocation-related behaviors[7,8], which is in alignment with dilute alloys. Those unexpected results observed in bulk HEAs may be attributed to the fact that the existence of unavoidable crystallographic defects, such as dislocations and vacancies, conceals the inherent elastic instability of HEAs. Reducing the sample size to nanoscale or even smaller could be an effective way to minimize pre-existing defects and to decipher the intrinsic lattice response to external stimuli[2,9–11].

In this work, the lattice evolution of nanosized HEA samples is investigated as a function of strain using in situ transmission electron microscopy (TEM). A typical method, in situ welding[12,13], is used for synthesizing nanoscale HEA samples with a dog-bone shape.

[1]Center for X-mechanics, School of Aeronautics and Astronautics, Zhejiang University, Hangzhou 310027, China. [2]ZJU-Hangzhou Global Scientific and Technological Innovation Center, Zhejiang University, Hangzhou 310027, China. [3]Beijing Advanced Innovation Center for Materials Genome Engineering, State Key Laboratory for Advanced Metals and Materials, University of Science and Technology Beijing, Beijing 100083, China. [4]Institute for Materials Intelligent Technology, Liaoning Academy of Materials, Shenyang 110010, China. [5]College of Materials Science and Engineering, Hunan University, Changsha 410082, China. [6]Shanghai Key Laboratory of Mechanics in Energy Engineering, Shanghai Institute of Applied Mathematics and Mechanics, School of Mechanics and Engineering Science, Shanghai University, Shanghai 200444, China. [7]State Key Laboratory of Silicon and Advanced Semiconductor Materials, School of Materials Science and Engineering, Zhejiang University, Hangzhou 310027, China. [8]Department of Materials Science and Engineering, University of California, Berkeley, CA 94720, USA. [9]These authors contributed equally: Yeqiang Bu, Yuan Wu. ✉e-mail: wuyuan@ustb.edu.cn; liujiabin@zju.edu.cn; htw@zju.edu.cn; luzp@ustb.edu.cn

The nearly defect-free HEAs fabricated are desirable samples for in situ atomic resolution characterization and mechanical testing. Using these nanoscale HEAs, we observe a distinct manifestation of elastic instability termed "elastic strain-induced amorphization", thereby broadening the conventional understanding beyond the classic framework of dislocation-mediated incipient plasticity. We further find that the local lattice periodicity vanishes as the elastic instability occurs at an elastic strain of ∽10%, which intrinsically differs from the amorphization process induced by defect accumulation. The latter has been reported in ordered TiNi alloys, covalent materials and most recently bulk HEAs[14–20]. Our findings offer insights into the understanding of elastic instability and atomistic mechanisms for basic physical and mechanical phenomena such as elastic instability and incipient plasticity.

## Results and Discussions

### Elastic strain-induced amorphization in nanoscale TiHfZrNb

Figure 1a shows a quaternary TiHfZrNb HEA sample prepared by in situ welding (details can be found in Methods). According to the high-resolution TEM images and the corresponding fast Fourier transform (FFT) images (see insets), it is clear that the initial lattice exhibits a typical BCC structure with a lattice constant of $a = b = c = 3.53$ Å. Uniaxial tensile loading along the [001] direction was applied to this HEA sample. The critical lattice structure right before the onset of elastic instability was captured by the in situ high-resolution TEM image in Fig. 1b1; the local lattice in the region outlined by the yellow square deviates from the BCC structure due to the local elastic strain. From the enlarged view and corresponding FFT pattern (Fig. 1b2), the lattice was deformed to a low-symmetry body-centered tetragonal (BCT) lattice with lattice constants of $a = b' = 3.42$ Å and $c' = 3.87$ Å. Figure 1b3 shows a simulated high-resolution image and the corresponding FFT image of the BCT lattice, which agrees well with the experimental data of Fig. 1b2. According to the lattice change, it can be determined that

the initial BCC lattice was elongated ∽10% along the $c$ axis (i.e., [001]), accompanied with a concomitant biaxial lateral contraction of ∽3% along the [100] and [010] directions, as schematically shown in Fig. 1b4. Neither lattice shift nor bond breakage is observed. The image implies the absence of inelastic energy relaxation, namely the 10% of deformation is elastic (rather than pseudoelasticity caused by phase transformation). The Poisson's ratio of TiHfZrNb calculated here is ∽0.3, which is similar to the previous theoretical results[21]. The loading-unloading during tension of the TiHfZrNb sample clearly verified that the large elastic strain can be completely recovered upon unloading (Supplementary Fig. 1). Such substantial elastic recovery strongly indicates that the absence of inelastic relaxation in the nanoscale HEA sample, including defects generation, even at such a high strain. This observation further verifies that dislocation generation is minimal or absent throughout the entire tensile process.

With further tension, the periodic crystalline lattice was disrupted, and a transition from BCT to amorphous structure occurred, as shown in the red square region of Fig. 1c1. Note that the free surface serves as the nucleation site for such elastic instability. Figure 1c2 is the corresponding enlarged image, showing disordered regions (outlined by the red dotted line) with the highly distorted lattice (enclosed by the yellow and red dotted lines) that deviates from the initial BCC lattice in front of the crystal-amorphous interface. During further tensile loading, the amorphous region rapidly spread across the section of the sample, as shown in Fig. 1d1. The corresponding FFT image in the inset of Fig. 1d1 shows a diffuse ring, indicative of an amorphous structure. The magnified image in Fig. 1d2 reveals the diffusive characteristics of the interface between crystalline and amorphous material, which possesses a transition layer of ∽1.5 nm containing a low-symmetry highly distorted lattice; this is outlined in the figure in red dotted lines. Such a wide and diffusive transition layer absorbs a large amount of elastic strain energy produced by interfacial coordination and acts as the stimulus for the growth of the amorphous structure.

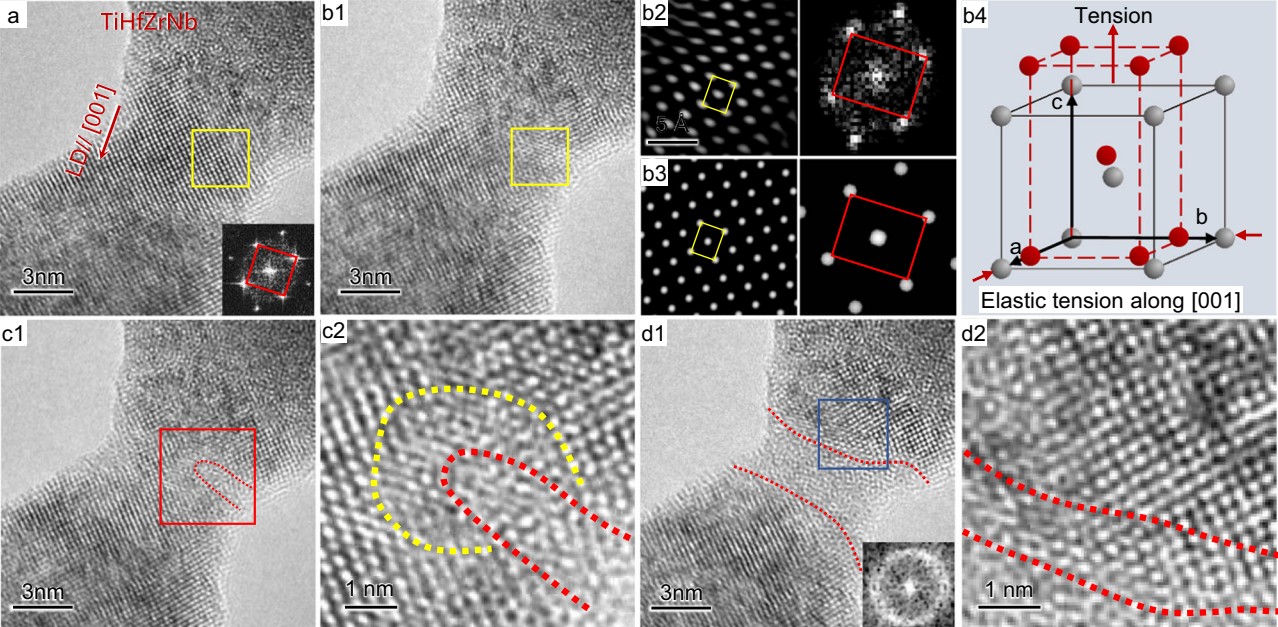

**Fig. 1 | Elastic strain-induced amorphization in TiHfZrNb samples. a** The initial HEA sample with body-centered cubic (BCC) lattice was stretched along [001]; the inset is a fast Fourier transform (FFT) image corresponding to the yellow squared region. **b1**, The critical frame before the onset of amorphization. **b2** The enlarged view and FFT of the yellow squared region in **b1**, indicating that the BCC lattice was elastically elongated to a body-centered tetragonal (BCT) lattice. **b3**, Simulated high-resolution transmission electron microscopy (TEM) image and the FFT pattern corresponding to **b2**. **b4** A schematic image illustrates the lattice deformation. **c1** An amorphous nucleus formed in the HEA sample. **c2**, The enlarged view corresponding to the red squared region in **c1**, the region outlined by red dotted line is the amorphous nucleus, and the region enclosed by yellow and red dotted lines is the highly distorted lattice in front of the crystal-amorphous interface. **d1** An amorphous segment formed in the HEA sample. **d2** The enlarged view corresponding to the blue squared region in **d1** shows a diffuse interface (outlined by two red dotted lines). LD is the abbreviation of loading direction.

The detailed in situ amorphization process is imaged in Supplementary Movie 1. Such mechanical-driven amorphization was further verified for tensile loading along other crystallographic directions, such as <110> directions of the TiHfZrNb samples, as shown in Supplementary Fig. 2. Similarly, the mechanical-driven amorphization took place at a critical elastic strain of ⌣10%. It is important to clarify that the thermal and damaging effects of irradiation in the electron beam of the TEM were negligible; as shown in Supplementary Fig. 3 and 4 in Supplementary Discussion 1 of the Supplementary Materials. Additionally, by tilting the sample to different angles, we confirmed that the observed disordered TEM contrast of the amorphous structure is not an artifact due to the viewing direction, as shown in Supplementary Fig. 5. It is worth mentioning that, although local lattice strain has been assessed by measuring variations in lattice spacing, the lack of in situ tensile curves limits our understanding of the role of such mechanical-driven amorphization in the mechanical response of nanoscale alloys. To address these challenges, future work should prioritize integrating the capabilities of 0.1 nm precision in displacement loading and nanonewton level force measurement accuracy into in situ TEM mechanical stage.

Mechanical-driven amorphization usually occurs due to significant accumulation of lattice defects, *e.g.*, dislocations, twins or stacking-fault tetrahedra, during plastic deformation in crystalline materials such as ordered TiNi alloy, covalent materials and recently reported in a bulk CrMnFeCoNi HEAs[14–20]. In contrast to this, only elastic stretching of bonds, with no generation of defects, was observed in the TiHfZrNb HEA samples before the onset of amorphization. The topological order was abruptly lost, and the amorphization was instantaneously activated as the elastic strain approached ⌣10%. This mechanically-driven disordering observed here is a manifestation of elastic instability, fundamentally different from the defect accumulation amorphization previously reported, and is herein referred to as elastic strain-induced amorphization.

The discovery of the elastic strain-induced amorphization phenomenon is attributed to the large elastic strain (⌣10%) in the nanosized HEA samples. The elastic limit in bulk samples, however, is usually less than 2% due to the ease of dislocation gliding. Although traditional defect accumulation-induced amorphization may occur in severely plastically deformed bulk samples—as evidenced by features such as amorphous bands in a 90% rolled bulk TiHfZrNb sample (as illustrated in Supplementary Discussion 2 and Supplementary Fig. 6 and 7), the elastic strain-induced amorphization cannot be observed in the bulk sample due to premature dislocation generation. Thus, we intentionally chose to use small-scale samples based on their intrinsic difficulties in defect premature nucleation, which is a necessary condition for triggering the present lattice instability behavior (i.e., elastic strain-induced amorphization).

## Mechanical responses for different nanoscale alloys

To check if such elastic strain-induced amorphization is a universal phenomenon in HEAs, the mechanical response from alloy samples with different numbers of principal elements were studied. In situ, high-resolution TEM was utilized to capture the differences in the mechanical-driven evolution of the lattice in these samples. The results of such experiments for the quinary TiHfZrNbTa, ternary TiZrNb, and binary ZrNb samples are shown in Fig. 2. The quinary TiHfZrNbTa sample shows a clear elastic strain-induced amorphization process under tension, similar to the TiHfZrNb (Figs. 2a1-3). As shown in Fig. 2a1, the sample before loading in tension displays a BCC lattice viewed along [001]. However, Fig. 2a2 and the inset reveal that this critical lattice structure only exists before the occurrence of elastic strain-induced amorphization under tension loading (around the [110] direction) when the BCC lattice is elongated into a distorted BCT lattice, and the lattice spacing of the (110) plane changes from 2.41 Å to 2.61 Å. The amorphous structure, produced by the elastic strain-induced amorphization and imaged in Fig. 2a3, occurs when the elastic strain exceeds a critical value of ⌣10%; this strain was determined from the change in lattice spacing of the (110) plane (Fig. 2a2). In the ternary TiZrNb sample, such an elastic strain-induced amorphization process was also observed at a critical elastic strain of ⌣10%, in this case, measured by the change in lattice spacing of the (200) plane, as shown in Figs. 2b1-b3. However, in the binary ZrNb sample (Figs. 2c1-c3), elastic strain-induced amorphization was not detected; instead, a mechanically-driven transformation to a face-centered cubic (FCC) phase occurred with a sharp BCC-FCC interface, as shown in Fig. 2c2. The orientation relationship of the parent BCC phase and the resultant FCC phase was determined to be $[111]_{BCC}//[110]_{FCC}$, $(\overline{1}10)_{BCC}//(\overline{1}11)_{FCC}$, which pertains to the well-known Kurdjumov-Sachs (K-S) relationship[22]. Figure 2c3 shows an FCC lattice projection along the [110] zone axis, from which the sample was found to retain its crystalline lattice fringes even after fracture. In Supplementary Fig. 8, the critical state of the elastically deformed lattice is shown before the crystalline transformation occurs. From the change in lattice spacing of the (110) plane, the critical elastic strain at transformation can be determined to be ⌣4%, which is significantly lower than the value to activate elastic strain-induced amorphization.

The in situ tension results under high-resolution TEM for the other nanosized alloys, specifically TaHfZrNb, HfZrNb, TiHfNb, TiNb, and TiZr, are given in Supplementary Fig. 9 and 10; the corresponding results are listed in Supplementary Table 1. To summarize, the elastic strain-induced amorphization phenomenon was observed in the ternary, quaternary, and quinary alloys at a lattice strain of ⌣10%, irrespective of loading directions. For the binary alloys, however, no elastic strain-induced amorphization could be observed; and dislocation slip and crystalline phase transformations prevailed instead. For the elemental metals, such as Nb, multiple modes of plastic deformation behavior in the crystalline regime has been reported, including dislocation slip, twinning, and crystalline phase transformation, but no elastic strain-induced amorphization could be activated in the nanosized samples[23]. Thus, the activation of elastic strain-induced amorphization generally depends on the total number of principal elements, *i.e.*, the more principal elements in the alloy, the more likely the amorphization can be triggered. Additionally, it is conceivable that the occurrence of elastic strain-induced amorphization is not solely attributed to size effects. As shown in Fig. 2c and Supplementary Fig. 10, the elastic instability observed in nanosized elemental and binary metals is characterized by dislocation behaviors, rather than the elastic strain-induced amorphization.

In addition to the number of principal elements, the influence of lattice structure on activating elastic strain-induced amorphization has also been taken into consideration. Nanosized FCC-structured FeCoNiCrCu and FeCoNiCrMn were stretched and their lattice evolution were dynamically observed (Supplementary Fig. 11), respectively, in which elastic strain-induced amorphization is absent, but dislocation motion mediates the elastic instability. The reasons for the difficult activation of low-temperature amorphization in FCC crystals have been analyzed by Greer[24], who pointed that close-packed structure and lower vibrational entropy of FCC structure hold the keys, as demonstrated in detail in Supplementary Fig. 12. Additionally, dislocation activation in BCC crystals needs to overcome a higher stress barrier due to their non-close-packed lattice structure[25–28], especially for BCC HEAs, which possess a more severe lattice distortion than FCC HEAs and dilute alloys[5]. Therefore, elastic strain-induced amorphization tends to occur in nanosized BCC-structured alloys with multiple principal elements.

As described in Supplementary Discussion 3 and Supplementary Fig. 13, the elastic strain-induced amorphization in BCC-structure HEAs was analyzed energetically using first-principle calculations (details of the methodology are given in the Methods section). Our observations clearly demonstrate that the energy barrier between the crystalline

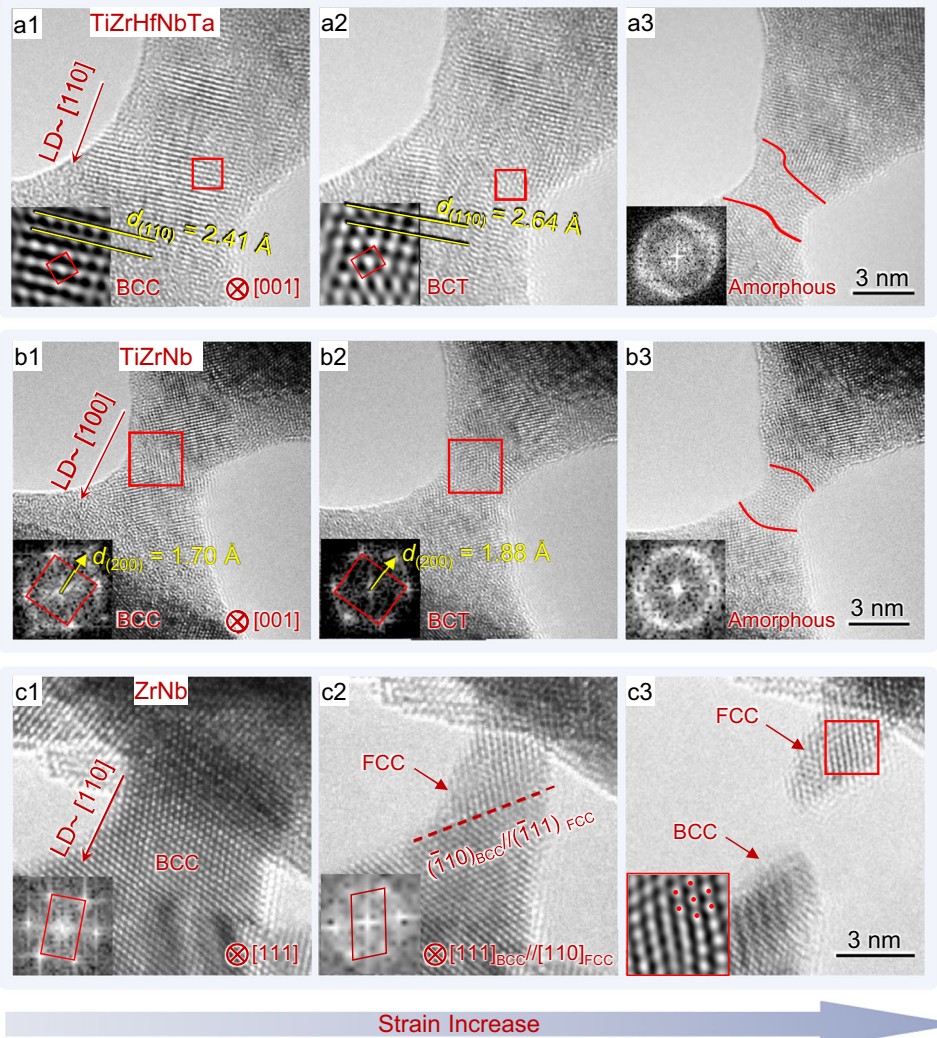

**Fig. 2 | Strain-induced lattice evolution of various samples with a different number of principal elements. a1–3** The elastic strain-induced amorphization in the quinary TiHfZrNbTa sample, and the insets in **a1–3** show the initial BCC lattice, the BCT lattice before the onset of the amorphization, and the FFT of the produced disordered structure, respectively. **b1–3** The elastic strain-induced amorphization in the ternary TiZrNb alloy sample, and the insets in **b1-3** show the FFT of the initial BCC lattice, the BCT lattice before the onset of the amorphization, and the FFT of the amorphous segment. **c1–3** The crystalline phase transformation from BCC to face-centered cubic (FCC) in the binary ZrNb sample, and the insets in **c1–3** show the FFT of the initial BCC lattice, the resulting FCC lattice and an enlarged view of the fractured sample, respectively. The FCC lattice is retained after fracture.

and the amorphous structures in the samples can be overcome by applying elastic strain energy provided a large (~10%) elastic strain can be achieved. In the alloy samples where the number of principal elements exceeds two, elastic strain-induced amorphization can occur once the elastic lattice strain reaches ~10% without the formation of defects such as dislocations or twins. However, in dilute alloys, such as the binary ZrNb alloy, displacive phase transformation via cooperative lattice shearing was found to occur at a lattice strain of 4%, with a release of elastic strain energy long before the energy barrier between the crystalline and amorphous structures was reached. Accordingly, it is apparent that, as in bulk alloys or nanosized dilute metals[23,29,30], due to the premature inelastic relaxation from the generation and propagation of lattice defects such as dislocations, twins and crystalline phase transformation, the critical elastic strain for amorphization in the binary ZrNb alloy hardly be attained.

### Origins of elastic strain-induced amorphization

To discover the origins of the suppression of dislocations, in HEA samples, differences in the lattices for the dilute metals and HEAs were investigated at an atomic scale using Density Functional Theory simulations. Elemental Nb and quaternary TiHfZrNb supercells containing 250 random-distributed atoms were constructed and fully relaxed to minimize the interatomic Hellmann-Feynman force[31]. The equilibrium lattices of Nb and TiHfZrNb are shown in Fig. 3 (a1 and a2), respectively. It is clear that the atoms in Nb strictly locate at the BCC lattice sites while those in TiHfZrNb considerably deviate from the ideal BCC lattice sites. As shown in Fig. 3b1, the radical distribution function (RDF) curve of the elemental Nb lattice is completely ordered, with the corresponding definitive peaks at the locations of the first-nearest-neighbor (1NN), the second-nearest-neighbor (2NN) and the third-nearest-neighbor (3NN) atomic pairs at 2.875, 3.325 and 4.675 Å, respectively.

For the equilibrium lattice of TiHfZrNb, conversely, the radial distribution is significantly widened (Fig. 3b2), confirming that the atoms significantly deviate from the regular lattice sites. In TiHfZrNb, the 2NN peak merges with the 1NN peak in the radial distribution function, and the distance between the 2NN and 3NN peaks is also smeared. The marked scattering of the RDF peaks is an indication of the significant lattice distortion in TiHfZrNb. The charge density distribution in the $(10\bar{1})$ cross-section plane was employed to visualize

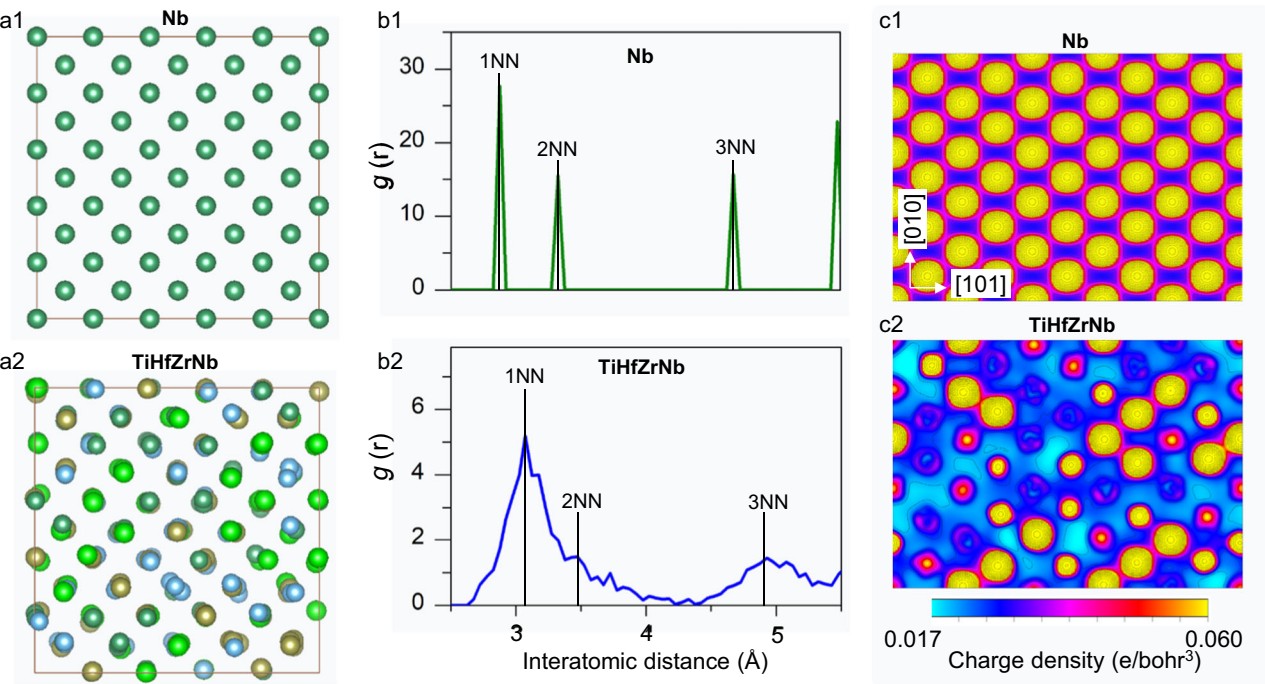

**Fig. 3 | The local inhomogeneity in HEAs. a1** and **2** The equilibrium lattice of Nb and TiHfZrNb. **b1** and **2** the radial distribution functions of Nb and TiHfZrNb, where 1NN, 2NN or 3NN represents the first-nearest-neighbors, the second-nearest- neighbors or the third-nearest-neighbors, respectively. **c1** and **2** The charge density distributions in the (10$\bar{1}$) cross-section planes of Nb and TiHfZrNb lattices.

the local atomic environment and interatomic potentials. The charge density distribution in Nb is completely homogeneous (Fig. 3c1), indicating identical local atomic environments. In contrast, the corresponding charge density distribution in TiHfZrNb is complex and somewhat disordered (Fig. 3c2), implying very inhomogeneous local atomic environments. The low-energy displacive pathway for atoms in these different local atomic regions will be significantly varied, which is the key to hamper the premature dislocation activation so as to achieve the large elastic strain. The consequent irregular short-range atomic displacements (as revealed by the molecular dynamic simulations in Supplementary Fig. 14) mediates the elastic instability at large elastic strains (e.g., ∽10%) in these locally inhomogeneous HEAs, and results in the elastic strain-induced amorphization.

Such local atomic inhomogeneity stems from the strong inhomogeneous chemical fluctuations in alloys with multiple principal elements, which has been revealed by the atomic-resolution energy-dispersive X-ray spectroscopy (EDX) mapping. Figure 4a is a high-angle annular dark field (HAADF)−scanning transmission electron microscopy (STEM) image of ternary TiZrNb alloy along [111] zone axis and the corresponding atomic-resolution EDX maps of the individual elements. Figures 4b1-b6 are zoomed-in images captured from the local regions in EDX maps, which identify the multiple local chemical groups (TiZr-, TiNb-, ZrNb-, Ti-, Zr- and Nb-rich clusters, respectively) in TiZrNb alloy. To quantify the atomic-scale chemical fluctuation, the line profiles of atomic fraction in the (1$\bar{1}$0) planes projected along the [111] direction were taken from the EDX maps (Fig. 4c). The line profiles indicate that the Ti, Zr and Nb elements all exhibit inhomogeneous fluctuations at the atomic scale, e.g., the atomic fraction of Ti reaches 62% but drops to 13% in the vicinity. Such inhomogeneous element distribution without an identified interface and well-defined size can be characterized as an incipient concentration wave, which can be analyzed by pair correlation functions of the atomic fraction for each element[32,33]. In general, a high and sharp peak of the pair correlation function at wavelength $r$ represents an incipient concentration wave with a characteristic period $r$, while a low and broad peak reflects that the element distribution is random. As plotted in Fig. 4d, a strong correlation peak for Ti and Nb at a concentration wavelength $r$ around 2 nm, and a similarly strong correlation peak for Zr at $r$ around 3 nm, indicate that the length scale of the incipient clustering in ternary TiZrNb is ∽2–3 nm. For comparison, the atomic-scale chemical distribution analysis has been performed in the binary TiNb alloy (Supplementary Fig. 15), where elements fluctuate mildly and are randomly distributed. Our results indicate that the inhomogeneous element distributions with salient features of incipient concentration waves extensively exist in alloys with multiple principal elements, which places a significant influence on the competition between dislocation generation and elastic strain-induced amorphization at the onset of elastic instability.

The difficulty in dislocation nucleation in nanoscale HEAs can be qualitatively accessed by the "hot spots" model, where atoms with relative displacements above a critical value (referred to as 'hot spots') are considered precursors to lattice defect nucleation[27,28]. The critical elastic shear strain ($\gamma$) derived from this model for dislocation nucleation can be expressed as follows:

$$\gamma = \frac{1}{A(T)}\left\{\left[\frac{\alpha}{b} - a_0\beta(\dot{\gamma},T)\right] \times \frac{1}{(\ln N)^{1/m}} - B(T)\right\} \qquad (1)$$

where, $A(T)$ and $B(T)$ are functions of temperature ($T$) only. $A(T)$ represents the atomic displacements along the straining direction while $B(T)$ accounts for the isotropic and random atomic displacements due to thermal agitation. $b$ is the magnitude of the Burgers vector. $\beta(\dot{\gamma},T)$ represents the additional thermal agitation component and is a function of the strain rate ($\dot{\gamma}$) and temperature. $\alpha$ and $m$ are constants related to the critical displacement value and the probability distribution of atomic displacements in the crystal, respectively. $N$ is the number of atoms in the efficient volume for defect nucleation, which directly links the critical elastic shear strain of dislocation nucleation to the sample size and packing factor of lattice. Namely, if the sample size or packing factor is increased, the first term in curved

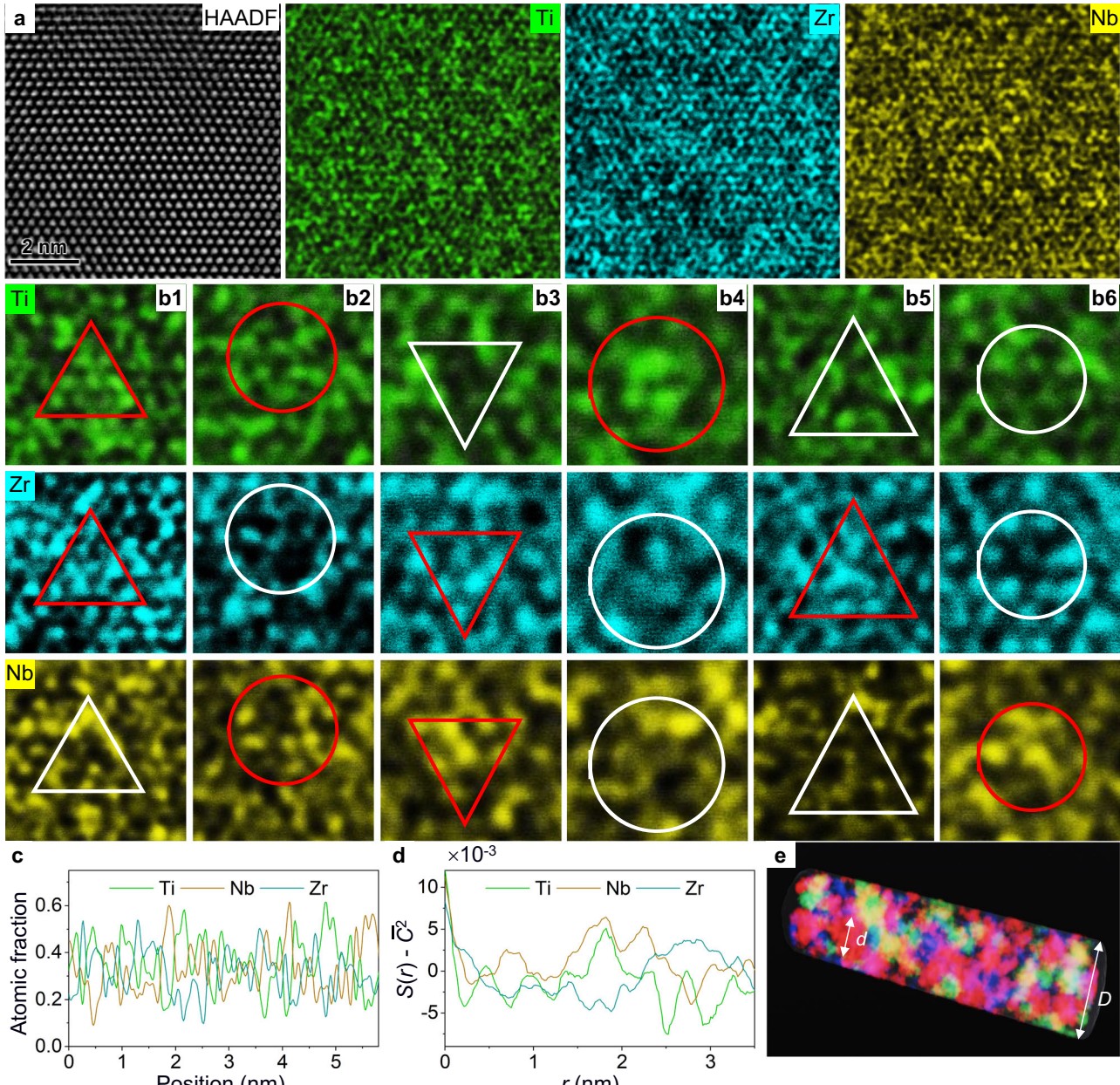

**Fig. 4 | Atomic-scale element distribution analysis for ternary TiZrNb alloys.** **a** HAADF-STEM image along the [111] zone axis, and corresponding atomically resolved EDX maps for individual elements of Ti, Nb and Zr. **b1–6** Zoomed-in images captured from the local regions in **a**, showing the (Ti, Zr)-, (Ti, Nb)-, (Nb, Zr)-, Ti-, Zr-, Nb-rich local groups in TiZrNb alloy, respectively. **c** Line profiles representing the distribution of individual elements in (1$\bar{1}$0) planes projected along the [111] direction. **d** The pair correlation functions ($S(r)$) of individual elements, where $r$ is concentration wavelength, $S(r)$ for each elements is shifted by the square of the average atomic fraction ($\bar{C}^2$), the detailed calculation methods for $S(r)$ follows the procedures in ref. 34. **e** Schematic illustration of the nanoscale HEA sample being divided into a series of domains due to local inhomogeneity.

brackets of Eq. (1) would decrease, and the critical strain should decrease, and vice versa.

In nanoscale HEAs characterized by chemical short-range ordering due to extensive atomic-scale heterogeneities, the volume of dislocation nucleation is further subdivided, as schematically shown in Fig. 4e (where $D$ and $d$ represent the diameter of the sample and the efficient volume for dislocation nucleation, respectively). Consequently, $N$ is dramatically decreased (even compared with that in nanoscale dilute alloys), and thus the critical elastic shear strain of dislocation nucleation ($\gamma$) is high in the nanoscale HEAs. On the other hand, due to the non-close-packed lattice structure, the packing factor of BCC lattices is lower than that of FCC lattices, indicating that BCC lattices usually possess fewer atoms per unit volume and

correspondingly higher resistance to dislocation nucleation. As a result, even at a high elastic strain, dislocation nucleation does not take place in nanoscale BCC HEAs, and eventually the elastic strain-induced amorphization occurred.

In summary, we have employed in situ high-resolution transmission electron microscopy to investigate the evolution of structure in complex concentrated high-entropy alloy samples. A manifestation of elastic instability, referred to as elastic strain-induced amorphization, was observed when the local elastic strain approached ~10%. The experimental results show that the activation of elastic strain-induced amorphization generally depends on the configuration entropy, specifically the total number of principal elemental constituents. The considerable local atomic inhomogeneity of HEA lattices, which

inhibits the activation of plastic deformation due to the motion of defects, such as dislocation slip and twinning, is deemed to be a major reason for the onset of elastic strain-induced amorphization. Our findings shed insights into the response of lattices to external stimuli and contribute to the fundamental understanding of elastic instability and incipient plasticity.

## Methods

### Sample preparation
All alloys used in this study were prepared by melting a mixture of pure metals with purity above 99.9 *wt.%* in a vacuum arc furnace. The ingots were remelted at least eight times to ensure chemical homogeneity. The samples used in this study, with a grain size ranging from ∽10 μm to 100 μm, possess a single phase and exhibit chemical homogeneity, as illustrated by the EBSD orientation map, EBSD phase map, and EDS map shown in Supplementary Fig. 16. The prepared ingots were sliced into rods with dimensions of 0.25 mm × 0.25 mm × 10 mm to fit into the TEM sample stage. The rods were torn apart with two clamps and mounted in the TEM sample stage in a glove box with a protective atmosphere of argon gas and then quickly transferred into TEM in a high-vacuum environment. The exposure time in the air of the samples with fresh nanotips was less than 10 s to minimize contamination. The TiHfZrNb sheet samples, each with a 3 mm thickness intended for macroscopic roll processing, were initially cut from the as-cast ingots. These samples subsequently underwent room-temperature rolling with reduction ratios of 30% and 90%. The rolled samples were then ground to approximately 50 μm in thickness and subjected to further chemical thinning through twin-jet electropolishing for subsequent TEM characterization.

### In situ TEM characterization
We prepared a series of nanosized samples by in situ welding, a technique that has been widely used for preparing samples in in situ high-resolution TEM experiments[12,13]. The in situ samples, prepared through in situ welding, exhibit a nearly circular cross-section with a diameter below 100 nm. A typical geometric characterization of the in situ sample is depicted in Supplementary Fig. 17. Here we utilized a high-resolution TEM equipped with a homemade X-Nano sample holder (developed by the Center for X-Mechanics, Zhejiang University[34]). As schematically shown in Supplementary Fig. 18, a square electric pulse generated by a signal pulse generator was applied on two nanotips obtained by tearing apart corresponding rods inside a JEOL 2100 F TEM. The samples were prepared under a square electric pulse with a duration time of 1 ms under a voltage range of 0-2 V. Driven by the nano-manipulator, the prepared sample was quasi-statically stretched at a strain rate of ∽10$^{-3}$ s$^{-1}$. Microstructure evolution was recorded by a charge-coupled device (CCD) camera at 2 frames per second.

### Atomic-resolution EDX mapping
Atomic-resolution quantitative EDX mappings were performed inside an aberration-corrected FEI Themis Z STEM operated at 300 kV with a spatial resolution of 60 pm. A Super-X EDX with four windowless silicon-drift detectors and a large collection solid angle of 0.7 steradians was equipped inside the STEM for atomic-resolution chemical analysis. The serviced EDX possesses an in-hole performance of <1% hole counts and low system background (<1% spurious peaks). The samples for quantitative chemical mappings with atomic resolution were fabricated by focus ion beam (FEI Helios 5 CX DualBeam) and further thinned to ∽30 nm (empirical estimate) by low-energy argon-ion milling (Fischione Model 1040 NanoMill). The thin sample was put into the microscope 12 hours before the characterization started to avoid possible mechanical drifting. The count rate was in the range of 200 to 500 counts per second when acquiring atomic-resolution EDX maps. To achieve a high signal-to-noise ratio, more than 1000 EDX mapping frames with the size of 512×512 pixels were collected using drift-corrected frame integration technology with a dwell time of 10 μs per pixel.

### First-principles calculations
Atomistic simulations were performed using Density Functional Theory (DFT) employing the Vienna ab initio simulation package within the projector augmented wave pseudopotential method. Each of the supercells constructed in this work contained 250 random-distributed atoms in equimolar ratios and with periodic boundary condition. The cutoff energy for the plane wave used was 300 eV; the system relaxation stops until the total energy converged to within 10$^{-5}$ eV atom$^{-1}$. The Monkhorst-Pack K-point meshes employed were 2×2×2. The energies of the tension-loaded supercells were calculated by the affine tensile deformation via the ADAIS code[35].

### Molecular dynamics (MD) simulations
The embedded-atom-method (EAM) empirical potential developed by Mishra[36] was used to calculate the interatomic interaction forces for the BCC TaNbHfZr alloy and Nb metal. Simulation boxes for both samples of $30a_0 \times 30a_0 \times 60a_0$ with BCC structure was built ($a_0$ is the lattice constant with the value of 3.54 Å for BCC TaNbHfZr and 3.30 Å for Nb). The gauge length, with a diameter of $15a_0$ and a height of $10a_0$, was formed by deleting atoms from the boxes and relaxing the cut-surfaces. Periodic boundary conditions were adopted for all three dimensions. Prior to initiating the tensile loading, the boxes were equilibrated at 300 K with the isothermal–isobaric (NPT) ensemble. Tensile deformation at a constant strain rate of $5 \times 10^8 \, s^{-1}$ was applied along the [001] direction. The MD simulations were carried out using the LAMMPS software. The common neighbor analysis (CNA) was used to discern whether an atom is in the FCC, BCC, HCP or amorphous structure. The analysis of CNA, atomic strain, and image visualization were performed using the OVITO software.

## Data availability
All data that support the key findings in this study are available within the main text and the Supplementary Information file. Additional raw data are available from the corresponding authors upon request. Source data are provided with this paper.

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

## Acknowledgements

This work was supported by the National Natural Science Foundation of China (grant no. 92266202 to J.B.L., grant nos. 52090022, 52288102 to H.T.W., grant no. 51921001 to Z.P.L. grant no. 52225103 to Y.W., grant no. 52122408 to H.H.W. and grant no. 12202379 to Y.Q.B.), the National Key Research and Development Program of China (grant no. 2022YFB4602101 to Y.W. and grant no. 2023YFB3712701 to Y.Q.B.), the Zhejiang Provincial Natural Science Foundation of China (grant no. LD24E010006 to J.B.L.), 111 Project (B07003 to Z.P.L.).

## Author contributions

Y.W., J.B.L., H.T.W., and Z.P.L. initiated the project. Z.F.L., Y.X.Y., and Y.W. prepared the HEA samples. Y.Q.B. and J.B.L. conducted the TEM characterization and corresponding data analysis. Y.Q.B. and P.W. performed the first-principal calculations. L.Q.L, X.J.L., and H.H.W carried out the MD simulations. Y.Q.B., Z.F.L., J.B.L., Y.W. R.O.R., Z.P.L., and W.Y. wrote the manuscript and all the authors contributed to the discussion and revision of the manuscript.

## Competing interests

The authors declare no competing interests.
