## [Peer Review File · Nature Communications]

Elastic strain-induced amorphization in high-entropy alloysREVIEWER COMMENTS

Reviewer #1 (Remarks to the Author):

The well-known Lindemann criterion indicates that melting in crystalline solids might occur when the root mean square of the atomic thermal displacements exceeds $\sim 10\%$ of the distance between the nearest-neighbor atoms, due to the advent of elastic instability with one of shear moduli vanishing. However, this criterion was never well proved experimentally due to the difficulty and limitation of experimental skills. In addition, almost all the metals and alloys will nucleate defects, for example the dislocations, and yield before the elastic deformation reaches $\sim 10\%$. This might be reason why most amorphization (or melting in this work) during plastic deformation is through the interaction between defects, since a lot of defects will be nucleated and interact with each other before the deformation reaches the critical value for amorphization or melting.

This article, however, obtained an elastic deformation up to $\sim 10\%$ by choosing BCC structured RHEAs in TEM through in-situ deformation. Both the methods and results were clearly presented. By using the experimental results and also the simulated data, the authors proved that amorphization depends on the configuration entropy, specifically the total number of principal elemental constituents. Personally, I think this is an important article and would like to recommend publication of the article in Nature Communications, since it might be the first article that experimentally authenticates the Lindeman criterion. On the other hand, I have two questions.

(1) The authors used in-situ TEM results to show the transformation from BCC to BCT and then to amorphous phase, and also indicated that there is no dislocation or very few dislocations generation during the whole tensile process. Do they have more solid proof to support this point? And do they have proofs to show that defects don't participate the amorphization process? This is very important, since all the conclusion in this article is based on this results.

(2) The authors indicated that the total number of principal elemental constituents has significant effect on the amorphization process. For example, the number should be larger than three. On the other hand, do the type of principal elemental constituents also has significant effect? And how the type affects the amorphization process?

Reviewer #2 (Remarks to the Author):

This paper uncovers the phenomenon termed "strain-induced melting," which serves as an empirical validation of the Lindemann criterion. The research in focus delves deep into the structural behavior of high entropy alloys (HEAs) under external stimuli such as mechanical load, employing a blend of in-situ high-resolution Transmission Electron Microscopy (TEM), selected area diffraction pattern (SADP), Density Functional Theory (DFT), and Molecular Dynamics (MD) simulations.

This phenomenon manifests when the elastic stability in certain HEAs reaches around 10%, leading to a transition from such as a Body-Centered Cubic (BCC) structure to a low-symmetry Body-Centered Tetragonal (BCT) and eventually to a disordered structure.

The research also reveals that the phenomenon's occurrence is influenced by the number of principal elements in the alloy and its lattice structure. Specifically, nanosized HEAs with multiple principal elements and a BCC lattice structure are more susceptible. DFT calculations strengthen these findings, indicating that a large elastic strain—approximately 10%—is sufficient to trigger this crystalline-to-amorphous transition.

Another key revelation is the suppression of dislocations in HEAs. Combining DFT and MD simulations, the study indicates that this unique mechanical property is due to the alloys' local atomic inhomogeneity and significant lattice distortion, which is further linked to strong chemical fluctuations in these multi-elemental alloys, a point supported by energy-dispersive X-ray spectroscopy (EDXS) mapping.

The study offers substantial contributions to the field of materials science, particularly in understanding the behavior of HEAs under mechanical stress. However, there are several notable areas where the paper could be strengthened:

1. **Initial Sample Characteristics:** The study lacks details on the initial characterization on elemental uniformity and microscopic features (such as phase and grain size) of the samples. Although some supplementary material provides information for certain samples, it's crucial to have this data for other samples as well.
2. **Size Effect:** While the paper does describe the preparation methods and shows TEM images, it fails to specify crucial details such as the shape, thickness, and dimensions of the in-situ samples. This omission makes it challenging to assess the repeatability and representativeness of the experiments, particularly critical for in-situ studies.
3. **MD Simulation Consistency:** The paper does not clarify whether the shapes of the samples used in Molecular Dynamics (MD) simulations (shown as 'dog-bone like' in Figure supplementary 14) are consistent with those used in the actual experiments (described as 'rod'). This could affect the validity of the simulation results.
4. **Mechanical Properties and Tensile Curves:** The absence of in-situ tensile curves is a significant gap. Such data would allow for a more direct comparison of mechanical behaviors among different samples, enhancing the study's comprehensiveness.
5. **Macroscopic Characteristics:** The paper lacks details on macroscopic characteristics like phase type for all mentioned samples and the bulk sample purity. For instance, the phase information and homogeneity/inhomogeneity of different bulk samples are not discussed. Terms like 'high purity' used in the context of in-situ experiments lack specificity. Detailed information is essential for a complete understanding of the material's behavior, especially in nanoscale.

Minor Issues:

1. Supplementary Figure 3 claims no structural change but doesn't provide evidence to substantiate this claim.
2. There's a lack of uniformity in sample preparation methods and nomenclature. For example, a 30% rolled sample is mentioned in supplementary materials but is not clearly stated the relationships and reasons in the main texts and methods part.
3. The paper makes an interesting comparison between small-scale 'order-to-disorder' transitions and macroscopic melting. Considering the study's focus on in-situ strain-induced 'melting,' incorporating macroscopic experiments, such as direct bulk tensile tests results comparison, would be a valuable addition.

Reviewer #3 (Remarks to the Author):

This manuscript reports the tensile deformation of high entropy alloy and correspondingly the structural evolution. The core conclusion of the work is the so-called discovery of deformation induced melting in high entropy alloy. The experimental evidence the authors provided is apparently a kind of solid state amorphization, which the authors expel on the basis of lack of defect activity. One of the fundamental questions for terming this as melting is the characteristics of the intrinsic product. Upon melting, the product should show liquid characteristics, which cannot be true or how to prove that. If this cannot be done, the work may dramatically mislead the community.

Further, the solid state amorphization as the team observed is essentially related to a small size effect of the very thin TEM sample. In order to prove a 10% elastic strain induced melting, the team could choose to deform a bulk sample up to 10%, then to see if they can observe similar thing, or even if a "melting" will happen. If this type of evidence cannot be provided, what contented in the present work is rather misleading.

In summary, I deeply believe what they observed belongs to solid state amorphization, with a condition of size effect. Terming this process as a melting is rather misleading to the community, as

they cannot prove it is indeed a liquid following the melting. Based on these high-level reasonings, I do not see any urgency of publication of this work.

Itemized Responses to the Reviewers' Comments

Reviewer #1 (Remarks to the Author):

The well-known Lindemann criterion indicates that melting in crystalline solids might occur when the root mean square of the atomic thermal displacements exceeds ~10% of the distance between the nearest-neighbor atoms, due to the advent of elastic instability with one of shear moduli vanishing. However, this criterion was never well proved experimentally due to the difficulty and limitation of experimental skills. In addition, almost all the metals and alloys will nucleate defects, for example the dislocations, and yield before the elastic deformation reaches ~10%. This might be reason why most amorphization (or melting in this work) during plastic deformation is through the interaction between defects, since a lot of defects will be nucleated and interact with each other before the deformation reaches the critical value for amorphization or melting.

This article, however, obtained an elastic deformation up to ~10% by choosing BCC structured RHEAs in TEM through in-situ deformation. Both the methods and results were clearly presented. By using the experimental results and also the simulated data, the authors proved that amorphization depends on the configuration entropy, specifically the total number of principal elemental constituents. Personally, I think this is an important article and would like to recommend publication of the article in Nature Communications, since it might be the first article that experimentally authenticates the Lindeman criterion. On the other hand, I have two questions.

(1) The authors used in-situ TEM results to show the transformation from BCC to BCT and then to amorphous phase, and also indicated that there is no dislocation or very few dislocations generation during the whole tensile process. Do they have more solid proof to support this point? And do they have proofs to show that defects don't participate the amorphization process? This is very important, since all the conclusion in this article is based on this results.

Reply: We thank the reviewer for this critical comment. *In situ* TEM is one of the most powerful approaches for revealing lattice evolution and defect generation at atomic resolutions^{1,2}. Herein, we employed *in situ* TEM to record the whole process of transition from the crystalline HEA to an amorphous structure at atomic resolution. We did not detect any dislocation generation during loading, and instead, only a severe elastic stretch of the BCC lattice occurred before the strain-induced melting (as shown in Fig. 1 in the main text).

To further verify the reliability of our conclusion regarding the absence of dislocations in the amorphization process, we conducted additional analyses from an alternative perspective, namely the loading-unloading process. As a typical example, Figure R1 (corresponding to Fig. S1 in Supporting Materials) showcases that a nanoscale HEA sample can be elastically stretched to a strain of 7.3% and fully recovers upon unloading. This substantial elastic recovery strongly indicates the absence of any inelastic relaxation (including defects generation), even at such high strains in the nanoscale HEA sample.

Figure R1. Tensile loading-unloading process of a nanosized TiHfZrNb sample. a, The initial TiHfZrNb sample; **b,** The TiHfZrNb sample is elastically stretched to 7.3% without any inelastic relaxation; **c,** The TiHfZrNb sample recovers its initial dimension after unloading.

To address reviewer's concern, we have incorporated the following sentences in the revised manuscript:

lines 9-13 on page 5

"Such substantial elastic recovery strongly indicates the absence of inelastic relaxation in the nanoscale HEA sample, including defects generation, even at such a high strain. This

observation further verifies that dislocation generation is minimal or absent throughout the entire tensile process."

(2) The authors indicated that the total number of principal elemental constituents has significant effect on the amorphization process. For example, the number should be larger than three. On the other hand, do the type of principal elemental constituents also has significant effect? And how the type affects the amorphization process?

Reply: Thank the reviewer for this thoughtful comment. The type of principal elemental constituents also has significant effects on the strain-induced melting. First, the strain-induced melting is absent in the FCC-structured HEAs constituted by 3d transition metal elements, which has been evidenced by *in situ* TEM characterization in Fig. S11 and discussed from the standpoint of Gibbs free energy in Fig. S12.

On the other hand, severe lattice distortion of HEAs is a key factor for occurrence of the strain-induced melting. In distorted lattices, there are very inhomogeneous local atomic environments (as revealed by charge density in Fig. 3), which hampers the dislocation activation and results in the strain-induced melting. It is conceivable that alloys with different element must possess a different degree of lattice distortion. Here, we estimated the lattice distortion degree in BCC-structured HEAs with different number and type of constituted elements by the following equation:

where Δd is adopted to describe the lattice distortion degree, and (x_i, y_i, z_i) and (x_i', y_i', z_i') are the coordinates of the unrelaxed and relaxed positions of atom i (the lattice was relaxed by the first principle calculations), respectively. The value of Δd of each alloy is plotted as a function of the number of components, as shown in Fig. R2 below. The lattice distortion does not increase monotonically with the number of constitute elements, which also affected by the specific type of constitutes. Nevertheless, ternary, quaternary and quinary alloys generally possess larger lattice distortion than binary alloys and pure metals, which seems to be a key factor for inducing strain-induced melting in the nanoscale alloys.

Figure R2. Lattice distortion degree of various alloys.

Reviewer #2 (Remarks to the Author):

This paper uncovers the phenomenon termed "strain-induced melting," which serves as an empirical validation of the Lindemann criterion. The research in focus delves deep into the structural behavior of high entropy alloys (HEAs) under external stimuli such as mechanical load, employing a blend of in-situ high-resolution Transmission Electron Microscopy (TEM), selected area diffraction pattern (SADP), Density Functional Theory (DFT), and Molecular Dynamics (MD) simulations.

This phenomenon manifests when the elastic stability in certain HEAs reaches around 10%, leading to a transition from such as a Body-Centered Cubic (BCC) structure to a low-symmetry Body-Centered Tetragonal (BCT) and eventually to a disordered structure.

The research also reveals that the phenomenon's occurrence is influenced by the number of principal elements in the alloy and its lattice structure. Specifically, nanosized HEAs with multiple principal elements and a BCC lattice structure are more susceptible. DFT calculations strengthen these findings, indicating that a large elastic strain—approximately 10%—is sufficient to trigger this crystalline-to-amorphous transition.

Another key revelation is the suppression of dislocations in HEAs. Combining DFT and MD simulations, the study indicates that this unique mechanical property is due to the alloys' local atomic inhomogeneity and significant lattice distortion, which is further linked to strong chemical fluctuations in these multi-elemental alloys, a point supported by energy-dispersive X-ray spectroscopy (EDXS) mapping.

The study offers substantial contributions to the field of materials science, particularly in understanding the behavior of HEAs under mechanical stress. However, there are several notable areas where the paper could be strengthened:

1. Initial Sample Characteristics: The study lacks details on the initial characterization on elemental uniformity and microscopic features (such as phase and grain size) of the samples. Although some supplementary material provides information for certain samples, it's crucial to have this data for other samples as well.

Reply: We thank the reviewer for these constructive comments. We have added initial

sample characterizations on their grain size, phase and elemental distribution, as shown in Fig. R3. The samples utilized in this study exhibit a grain size ranging from $\sim 10\ \mu\text{m}$ to $100\ \mu\text{m}$, as revealed by the EBSD orientation map. Furthermore, these samples demonstrate a single phase and showcase chemical homogeneity at the microscale, as indicated by the EBSD phase map and EDS map.

Figure R3. EBSD orientation map, EBSD phase map and EDS map for samples used in this study.

Following the reviewer's comments, we have added those initial sample characterizations as Fig. S15 in the Supplementary materials, along with the corresponding description in the Method part:

Supplementary Figure 16 | EBSD orientation map, EBSD phase map and EDS map for the samples used in this study."

" The samples used in this study, with a grain size ranging from $\sim 10 \mu\text{m}$ to $100 \mu\text{m}$, possess a single phase and exhibit chemical homogeneity, as illustrated by the EBSD orientation map, EBSD phase map and EDS map shown in Fig. S16."

2. Size Effect: While the paper does describe the preparation methods and shows TEM images, it fails to specify crucial details such as the shape, thickness, and dimensions of the in-situ samples. This omission makes it challenging to assess the repeatability and representativeness of the experiments, particularly critical for in-situ studies.

Reply: We thank the reviewer for these critical comments. In our study, we adopted the typical method, namely the *in situ* welding to prepare nanoscale HEA samples for *in situ* observations³⁻⁶. The samples prepared through *in situ* welding typically have a nearly circular cross-section with a diameter below 100 nm , a characteristic that has been previously discussed in other's earlier work³.

In response to the reviewer's comments, we have undertaken additional basic

characterization of the sample geometry, as illustrated in Fig. R3. The cross-sectional shape of the nanoscale HEA sample determined approximately by tilting the nanoscale sample along the α direction from -25° to $+25^\circ$. The diameter of the nanoscale sample changes little, indicating that the cross section of the nanoscale HEA sample is nearly circular.

Figure R4. Cross-section of the nanoscale HEA sample observed by tilting the nanoscale sample along the α direction from -25° to $+25^\circ$. The diameter of the nanoscale sample changes little, which indicates that the cross section of the nanoscale HEA sample is nearly circular.

To comply with the reviewer's comments, we have added statements on geometry of the *in situ* samples in Method part and Fig. S16 in the supplementary materials:

"The *in situ* samples, prepared through *in situ* welding, exhibit a nearly circular cross-section with a diameter below 100 nm. A typical geometric characterization of the *in situ* sample is depicted in Fig. S16."

Supplementary Figure 17 | Cross-section of the nanoscale HEA sample observed by tilting the nanoscale sample along the α direction from -25° to $+25^\circ$. The diameter of the nanoscale sample changes little, which indicates that the cross section of the nanoscale HEA sample is nearly circular."

3. MD Simulation Consistency: The paper does not clarify whether the shapes of the samples used in Molecular Dynamics (MD) simulations (shown as 'dog-bone like' in Figure supplementary 14) are consistent with those used in the actual experiments (described as 'rod'). This could affect the validity of the simulation results.

Reply: Thank the reviewer for the kind concern. The MD simulations were conducted with the aim of reproducing and elucidating the distinct mechanisms underlying elastic instability in elemental metals and HEAs, namely, dislocation slip and strain-induced melting, respectively, as observed experimentally. Notably, the MD simulations meticulously reproduce key experimental parameters, including nanosized sample dimensions, varying degrees of lattice distortion in the two samples, circular cross-sections, and uniaxial tensile loading, ensuring a faithful representation of the experimental conditions. Regarding the variation in sample shape, whether dog-bone like or rods, its influence on accurately simulating the stress state and eventual instability mechanism is actually minimal.

4. Mechanical Properties and Tensile Curves: The absence of in-situ tensile curves is a significant gap. Such data would allow for a more direct comparison of mechanical behaviors

among different samples, enhancing the study's comprehensiveness.

Reply: Thank the reviewer for these constructive comments. As the reviewer pointed out, *in situ* tensile curves can visually correlate structural evolution with mechanical performance. However, for the small-scale samples of only ~10 nm size in this study, measuring the *in situ* tensile curves requires instrument capabilities of 0.1 nm precision in displacement loading and nano newton-level force measurement accuracy. These instrument-related limitations for measuring tensile curves of nanoscale samples are currently the common challenge for the field. Therefore, we employed a method based on the variation in the lattice spacing derived from HRTEM images to calculate local lattice strain changes, which is a commonly used method for local strain measurements in *in situ* mechanical testing of small-sized samples^{3,4,7}.

5. Macroscopic Characteristics: The paper lacks details on macroscopic characteristics like phase type for all mentioned samples and the bulk sample purity. For instance, the phase information and homogeneity/inhomogeneity of different bulk samples are not discussed. Terms like 'high purity' used in the context of in-situ experiments lack specificity. Detailed information is essential for a complete understanding of the material's behavior, especially in nanoscale.

Reply: Thank the reviewer for these constructive comments.

Following the reviewer's inquiries, we have added the details on sample preparation (in Methods part) and initial characterization for all the alloys used in this study (Fig. S15):

"All alloys used in this study were prepared by melting a mixture of pure metals with purity above 99.9 wt.% in a vacuum arc furnace. The ingots were remelted at least eight times to ensure chemical homogeneity. The samples used in this study, with a grain size ranging from ~10 μm to 100 μm , possess a single phase and exhibit chemical homogeneity, as illustrated by the EBSD orientation map, EBSD phase map and EDS map shown in Fig. S15. Then, the prepared ingots were sliced into rods with dimensions of 0.25 mm \times 0.25 mm \times 10 mm to fit into the TEM sample stage."

Minor Issues:

1. Supplementary Figure 3 claims no structural change but doesn't provide evidence to

substantiate this claim.

Reply: Sorry for the missing details. We have incorporated additional descriptions in Supplementary Discussion 1 and provided more comprehensive details in Supplementary Figure 3:

"To evaluate the "knock-on" effects of electron irradiation on the nanoscale HEA samples, we exposed a nanoscale TiHfZrNb sample to an electron beam for 30 minutes. Subsequent examination of this sample revealed that the shape and location of crystal-amorphous interface, crystal orientation and lattice spacing in the TEM pictures and the corresponding FTT image (Fig. S3) remained unchanged after irradiation, indicating that the incident beam did not cause a "knock on" effect during this exposure period."

Supplementary Figure 3 | A nanoscale HEA sample was exposed to electron irradiation for 30 min. No structural change can be detected (a) before and (b) after irradiation. The crystal-amorphous interface was outlined by the yellow lines, and the insets are corresponding FTT images."

2. There's a lack of uniformity in sample preparation methods and nomenclature. For example, a 30% rolled sample is mentioned in supplementary materials but is not clearly stated the relationships and reasons in the main texts and methods part.

Reply: We thank the reviewer for this kind reminder. Characterization of the microstructure of the 30% rolled bulk TiHfZrNb alloy is to demonstrate the high density of dislocations generated prior to amorphization in the bulk alloys. As such, the mechanism underlying the generation of amorphous bands in the 90% rolled bulk TiHfZrNb alloy is attributed to defect accumulation-induced amorphization. To comply with the reviewer's

comments, we added the preparation details of the bulk rolled samples in the method part:

"The TiHfZrNb sheet samples, each with a 3 mm thickness intended for macroscopic roll processing, were initially cut from the as-cast ingots. Subsequently, these samples underwent room-temperature rolling with a reduction ratio of 30% and 90%. The rolled samples were then ground to approximately 50 μm in thickness and subjected to further chemical thinning through twin-jet electropolishing for subsequent TEM characterization."

3. The paper makes an interesting comparison between small-scale 'order-to-disorder' transitions and macroscopic melting. Considering the study's focus on in-situ strain-induced 'melting,' incorporating macroscopic experiments, such as direct bulk tensile tests results comparison, would be a valuable addition.

Reply: Thank the reviewer for the kind suggestion. The post-mortem microstructure characterization of the bulk TiHfZrNb sample after tensile tests has been performed in our previous work⁸. The TEM and XRD results (as shown in Fig. R4) show that dislocation slip mediates the deformation of bulk TiHfZrNb, whilst the strain-induced melting is absent. This absence is attributed to the low elastic strain limit characteristic of bulk alloys, typically less than 2%.

We have previously addressed this information in Supplementary Discussion 2, stating, "The post-mortem microstructure characterization of bulk TiHfZrNb samples after tensile tests has been performed in our previous work⁵. The TEM and XRD results show that dislocation slip mediates the deformation of bulk TiHfZrNb, while strain-induced melting is absent".

Figure R5. Dislocations in bulk TiHfZrNb samples pre-strained to (a) 2.5%, (b) 8% and (c) eventually fracture. (d) X-ray diffraction patterns for the as-cast, 5% tensioned, and fractured TiHfZrNb HEA samples, respectively. These data were taken our previously published work⁸.

Reviewer #3 (Remarks to the Author):

This manuscript reports the tensile deformation of high entropy alloy and correspondingly the structural evolution. The core conclusion of the work is the so-called discovery of deformation induced melting in high entropy alloy. The experimental evidence the authors provided is apparently a kind of solid state amorphization, which the authors expelled on the basis of lack of defect activity. One of the fundamental questions for terming this as melting is the characteristics of the intrinsic product. Upon melting, the product should show liquid characteristics, which cannot be true or how to prove that. If this cannot be done, the work may dramatically mislead the community.

Further, the solid state amorphization as the team observed is essentially related to a small size effect of the very thin TEM sample. In order to prove a 10% elastic strain induced melting, the team could choose to deform a bulk sample up to 10%, then to see if they can observe similar thing, or even if a “melting” will happen. If this type of evidence cannot be provided, what contented in the present work is rather misleading.

In summary, I deeply believe what they observed belongs to solid state amorphization, with a condition of size effect. Terming this process as a melting is rather misleading to the community, as they cannot prove it is indeed a liquid following the melting. Based on these high-level reasonings, I do not see any urgency of publication of this work.

Reply: We thank the reviewer for these critical comments.

1) Indeed, the “melting” concept here is a solid-state amorphization. Extensive studies have recognized the similarity between this solid-state amorphization process and the real melting. For examples, Cahn and Johnson pointed out the analogues which exists in both the processes involving the nucleation of disorder ⁹, whilst Okawa discussed the parallels in the volume dependence of the shear modulus during irradiation-induced amorphization and heat-induced melting ¹⁰. Subsequently, a special term, *i.e.*, **the inverse melting**, was employed for the nomenclature of this kind of solid-state amorphization during which a crystal reversibly

transforms into an amorphous phase (**regarded as a kind of liquid phase from an atomic packing point of view**) upon cooling, contrary to the melting on heating^{11,12}. Here, we inherit this term to describe the amorphization of our nanoscale HEA samples after very large elastic deformation, **which is a novel phase transformation mode and fundamentally different from the defect accumulation amorphization previously reported.**

2) **The occurrence of strain-induced melting is not solely attributed to size effects.** Instead, its underlying mechanism is rooted in the intrinsic local heterogeneity within the HEAs. As compelling evidences shown in Figs. 2c and S10, where elastic instability observed in nanosized elemental and binary metals is characterized by dislocation behaviors, rather than strain-induced melting.

In our experimental design, the deliberate selection of nanosized HEA samples serve to provide empirical validation for Lindemann's theory, as discussed in the Introduction section. The combination of a constrained sample volume and the intrinsic local heterogeneity within HEAs collaboratively suppresses the generation of premature dislocation generation during the loading process. This, in turn, facilitates the achievement of significant elastic strain required to initiate strain-induced melting, thus empirically verifying Lindemann's theory.

3) Furthermore, achieving a comparable elastic strain of ~10% in bulk HEA samples proves challenging. As illustrated by the typical tensile strain-stress curve of bulk TiHfZrNb samples (Fig. R6) and confirmed by numerous previous studies¹³⁻¹⁶, the elastic limit in bulk samples is usually less than 2% due to the ease of dislocation gliding. Therefore, although traditional defect accumulation-induced amorphization may occur in severely plastically deformed bulk samples—as evidenced by features such as amorphous bands in a 90% rolled bulk TiHfZrNb sample (as depicted in Fig. S6)—amorphization through strain-induced melting cannot be observed in bulk sample due to premature dislocation generation.

Figure R6. A typical tensile true stress-strain curve of a bulk TiHfZrNb sample⁸.

As pointed out by both Reviewer 1 and 2, our current research has made significant contributions to the field of materials science, particularly in understanding the behavior of HEAs under mechanical stress. In addition, our manuscript is an important article as it is the first to experimentally validate the Lindeman criterion. To comply with the reviewer's comments, nevertheless, we have added relative description of the nomenclature of the strain-induced 'melting', the clarification of size effect and the reasons for the absence of strain-induced 'melting' in bulk HEAs:

lines 2-10 on page 7

"Extensive studies have recognized the similarity between solid-state amorphization and actual melting processes⁴³⁻⁴⁶. Specifically, a special term (i.e., the inverse melting^{43,44}) was employed for the nomenclature of solid-state amorphization during which the crystalline phase reversibly transforms into an amorphous phase (regarded as a liquid phase from an atomic packing point of view) upon cooling, contrary to the melting upon heating. Here, we inherit this term to describe the amorphization of our nanoscale HEA samples after very large elastic deformation, termed strain-induced 'melting', which is a novel phase transformation mode fundamentally different from the defect accumulation amorphization previously reported."

lines 12-18 on page 7

"However, the elastic limit in bulk samples is usually less than 2% due to the ease of dislocation gliding. Therefore, although traditional defect accumulation-induced amorphization may occur in severely plastically deformed bulk samples—as evidenced by features such as

amorphous bands in a 90% rolled bulk TiHfZrNb sample (as illustrated in Supporting Discussion 2 and Figs. S6 and 7), amorphization through strain-induced melting cannot be observed in bulk sample due to premature dislocation generation."

lines 18-22 on page 9

" Additionally, it is conceivable that the occurrence of strain-induced melting is not solely attributed to size effects. As shown in Figs. 2c and S10, the elastic instability observed in nanosized elemental and binary metals is characterized by dislocation behaviors, rather than the strain-induced melting."

Referenece

- 1 Zheng H, Meng Y S, Zhu Y. Frontiers of in situ electron microscopy[J]. MRS Bulletin, 2015, 40(1): 12-18.
- 2 Zhang C, Firestein K L, Fernando J F S, et al. Recent progress of in situ transmission electron microscopy for energy materials[J]. Advanced Materials, 2020, 32(18): 1904094.
- 3 Wang J, Zeng Z, Weinberger C R, et al. In situ atomic-scale observation of twinning-dominated deformation in nanoscale body-centred cubic tungsten[J]. Nature materials, 2015, 14(6): 594-600.
- 4 Zheng H, Cao A, Weinberger C R, et al. Discrete plasticity in sub-10-nm-sized gold crystals[J]. Nature communications, 2010, 1(1): 144.
- 5 Zhu Q, Cao G, Wang J, et al. In situ atomistic observation of disconnection-mediated grain boundary migration[J]. Nature communications, 2019, 10(1): 156.
- 6 Zhong L, Wang J, Sheng H, et al. Formation of monatomic metallic glasses through ultrafast liquid quenching[J]. Nature, 2014, 512(7513): 177-180.
- 7 He Y, Zhong L, Fan F, et al. In situ observation of shear-driven amorphization in silicon crystals[J]. Nature nanotechnology, 2016, 11(10): 866-871.
- 8 Bu Y, Wu Y, Lei Z, et al. Local chemical fluctuation mediated ductility in body-centered-cubic high-entropy alloys[J]. Materials Today, 2021, 46: 28-34.
- 9 Cahn R W, Johnson W L. The nucleation of disorder[J]. Journal of Materials Research, 1986, 1(5): 724-732.
- 10 Ookawa A. On the mechanism of deformation twin in fcc crystal[J]. Journal of the Physical Society of Japan, 1957, 12(7): 825-825.
- 11 Greer A L. The thermodynamics of inverse melting[J]. Journal of the Less Common Metals, 1988, 140: 327-334.
- 12 Greer A L. Too hot to melt[J]. Nature, 2000, 404(6774): 134-135.
- 13 He Q F, Wang J G, Chen H A, et al. A highly distorted ultraelastic chemically complex Elinvar alloy[J]. Nature, 2022, 602(7896): 251-257.
- 14 Ding Q, Zhang Y, Chen X, et al. Tuning element distribution, structure and properties by composition in high-entropy alloys[J]. Nature, 2019, 574(7777): 223-227.
- 15 Lei Z, Liu X, Wu Y, et al. Enhanced strength and ductility in a high-entropy alloy via ordered oxygen complexes[J]. Nature, 2018, 563(7732): 546-550.
- 16 Wei S, Kim S J, Kang J, et al. Natural-mixing guided design of refractory high-entropy alloys with as-cast tensile ductility[J]. Nature Materials, 2020, 19(11): 1175-1181.

Reviewers' comments:

Reviewer #1 (Remarks to the Author):

The authors have well addressed all my comments, and also the comments from another reviewer. I recommend acceptance of the article by Nature Communications.

Reviewer #2 (Remarks to the Author):

The authors' responses address the major concerns raised in the review. They have provided additional data and clarifications that significantly enhance the paper's comprehensiveness and scientific rigor.

However, there are still two parts that could be further improved:

1. MD Simulation Consistency: The response is satisfactory. The authors have explained how the simulations align with experimental conditions.

Providing a more detailed justification for the minimal impact of shape variation would strengthen their argument.

Additionally, better clarifications for each term and clearer correlations between simulations and experiments will enhance the study.

2. Mechanical Properties and Tensile Curves: The response is reasonable, acknowledging technical limitations and offering an alternative approach. However, it would be more convincing if the authors could suggest potential future work to address these limitations.

Once these issues are resolved, I believe the paper will be ready for publication.

Reviewer #3 (Remarks to the Author):

I read the revised version very carefully and thanks to the authors for carefully addressing the comments of mine and other reviewers. Regarding the size effect, which is similarly asked by reviewer 2, it is apparent that they cannot duplicate at bulk state what they observed under in-situ TEM observation. They stated that this difference is due to low elastic strain of only 2% for the case of bulk state. While for thin foil of in-situ TEM, they can reach 10%. Apparently, at low dimension you can lead to large strain due to the less degree of confinement. This inconsistency really defies the validity of the conclusion. Large body of literatures also documented about defect activity with respect to size effect.

The authors still tone the work along the line of "melting", which they use the argument of Lindemann criterion to support what they try to claim. For melting, Lindemann criterion only serves as a necessary criterion, but not sufficient one.

Overall, even the authors addressed the reviewers comments, I do not believe what claimed to be true as melting, it is only a size related amorphization, as indeed they cannot duplicate in bulk what they see at a thin foil.

I do not want to prevent publication of the work, while I feel publication of this work will mislead the community along the way for a forgeable period.

Reviewer #4 (Remarks to the Author):

Overall, this reviewer also agrees with the opinion of Reviewer #3 that the observed phenomenon is

closer to solid-state amorphization than melting. The basis on which the authors claim the phenomenon as melting are: i) the mechanical strain reaches 10%, which is comparable to the Lindemann criterion for melting, ii) there were no defects accumulation that mediates the solid-state amorphization, and iii) occurrence of plasticity was suppressed due to the inhomogeneous nature of atomic structures in HEAs. However, none of these three pieces of evidence look sufficient to fully support the authors' claim for strain-induced melting of the HEAs. Here are the reasons:

1) As the authors mentioned in the introduction, the Lindemann criterion states that melting would occur when the root mean square of the atomic "thermal" displacement exceeds $\sim 10\%$. Given that the criterion is formulated in terms of root mean squared displacement, not by the average displacement, it usually assumes the stochastic atomic motion, giving zero net atomic displacements. However, the mechanical atomic displacements in this work are all aligned along the tensile direction, so it is unclear how the two distinct phenomena having different underlying assumptions can be combined.

2) Even though the authors claim there are no defects, e.g., dislocation, that can mediate the solid-state amorphization, the samples actually have them, i.e., the free surface. Given the very small sample size in this work, the authors may need to verify that the role of the free surface is negligible to mediate amorphization.

3) In the supplementary discussion, the authors claim that the electron beam irradiation is negligible because i) no knock-on damages were observed, and ii) the temperature rise is negligibly small. However, there is an additional effect the authors may need to consider, i.e., under electron beam irradiation, the atoms in the metallic materials get continuously ionized, which would alter the properties of the HEAs far from their ground-state values. The easiest way to verify the electron irradiation effects are negligible is by performing additional tensile experiments in the TEM with the electron beam off during tensile deformation. If the authors observe the same kinds of amorphization (or melting according to the authors' claim) with beam-off after $\sim 10\%$ tensile elongation, they can safely claim that there were no electron beam effects.

4) The authors' analysis of the dislocations is confusing. First, using the DFT calculations, they proved that the atomic structures in the HEAs are inhomogeneous (though this claim itself is somewhat obvious). However, they did give a reasonable explanation of how this structural inhomogeneity contributes to suppressing the dislocation plasticity. The authors' analysis based on the Frank-Read model does not seem to be appropriate here. Generally, in the nano-sized sample, there is not enough space to accommodate two pinning points to create the Frank-Read (FR) source, so usually, the FR source does not exist. Anyhow, the authors also claim there was no dislocation in their samples. The FR source is basically a dislocation segment with two pinning points. The authors would be better off focusing on the surface nucleation of dislocations and need to convincingly convey how the atomic-structural heterogeneity induces the suppression of dislocation nucleation.

Overall, the evidence on which the authors claimed strain-induced melting does not seem to be convincing enough. Instead, there is a risk of misleading the readers of this article.

Itemized Responses to the Reviewers' Comments

Reviewer #1 (Remarks to the Author):

The authors have well addressed all my comments, and also the comments from another reviewer. I recommend acceptance of the article by Nature Communications.

Authors' response:

We appreciate the positive responses from the reviewers and the recommendation for acceptance in Nature Communications. Additionally, we want to express our sincere thanks to the reviewer for their previous feedback, which has been very helpful in improving our manuscript.

Reviewer #2 (Remarks to the Author):

The authors' responses address the major concerns raised in the review. They have provided additional data and clarifications that significantly enhance the paper's comprehensiveness and scientific rigor.

However, there are still two parts that could be further improved:

1. MD Simulation Consistency: The response is satisfactory. The authors have explained how the simulations align with experimental conditions.

Providing a more detailed justification for the minimal impact of shape variation would strengthen their argument.

Additionally, better clarifications for each term and clearer correlations between simulations and experiments will enhance the study.

Authors' response:

We appreciate the reviewer for the constructive suggestions. The region of interest in our study is located in the middle of the sample, characterized by stress concentration, and exhibits a circular cross-section in both experimental and simulated conditions. This midpoint is far away from the specimen ends where external loads were applied. As a result, we can confidently disregard any deviation in stress distribution associated with variations in the shape of the specimen ends, as illustrated by St. Venant's principle [Mechanics of Materials. *John Wiley & Sons*, 2020].

We have added these descriptions in the caption of Fig. S14:

"Additionally, our research focuses on the middle position of the sample, which is characterized by stress concentration and exhibits a circular cross-section in both experimental and simulated conditions. This midpoint is far away from the specimen ends where external loads were applied. As a result, we can confidently disregard any deviation in stress distribution associated with variations in the shape of the specimen ends, as illustrated by St. Venant's principle⁷."

In addition to adding the explanation of the correlation between simulations and experiments, the terms of "affine displacement" and "non-affine displacement" used in simulation were also clearly clarified in the caption of Fig. S14:

"Affine displacement is a geometric transformation that linearly alters the position of atoms in space, reflecting the uniform movement of atoms along the slip plane facilitated by dislocation

slipping. Conversely, non-affine displacement captures the disordered movement of atoms during amorphization."

2. Mechanical Properties and Tensile Curves: The response is reasonable, acknowledging technical limitations and offering an alternative approach. However, it would be more convincing if the authors could suggest potential future work to address these limitations.

Authors' response:

We thank the reviewer for the constructive comment. We have added some perspectives on precise measurement of mechanical properties and tensile curves of such nanoscale samples on line 12-18, page 5 of the revised manuscript, as shown below:

" It is worth mentioning that, although local lattice strain has been assessed by measuring variations in lattice spacing, the lack of *in situ* tensile curves limits our understanding of the role of such mechanical-driven amorphization in the mechanical response of nanoscale alloys. To address these challenges, future work should prioritize integrating the capabilities of 0.1 nm precision in displacement loading and nanonewton level force measurement accuracy into *in situ* TEM mechanical stage."

Once these issues are resolved, I believe the paper will be ready for publication.

Authors' response:

Thank the reviewer for the positive response to our revised manuscript. We are grateful for the reviewer's constructive comments, which have played a crucial role in improving the comprehensiveness and scientific rigor of our manuscript.

Reviewer #3 (Remarks to the Author):

I read the revised version very carefully and thanks to the authors for carefully addressing the comments of mine and other reviewers. Regarding the size effect, which is similarly asked by reviewer 2, it is apparent that they cannot duplicate at bulk state what they observed under in-situ TEM observation. They stated that this difference is due to low elastic strain of only 2% for the case of bulk state. While for thin foil of in-situ TEM, they can reach 10%. Apparently, at low dimension you can lead to large strain due to the less degree of confinement. This inconsistency really defies the validity of the conclusion. Large body of literatures also documented about defect activity with respect to size effect.

The authors still tone the work along the line of “melting”, which they use the argument of Lindemann criterion to support what they try to claim. For melting, Lindemann criterion only serves as a necessary criterion, but not sufficient one.

Overall, even the authors addressed the reviewers' comments, I do not believe what claimed to be true as melting, it is only a size related amorphization, as indeed they cannot duplicate in bulk what they see at a thin foil.

I do not want to prevent publication of the work, while I feel publication of this work will mislead the community along the way for a forgeable period.

Authors' response:

We thank the reviewer for the thoughtful concern. The primary point of debate between us and the reviewer revolves around the appropriateness of characterizing this abnormal solid-state amorphization process as "melting". In light of several unwarranted misunderstandings of our nomenclature by reviewers, we have opted to replace the term "strain-induced melting" with "elastic strain-induced amorphization" to precisely describe the novel elastic instability that we have observed. We have also reshaped the Introduction part and placed the emphasis on revealing a new manifestation of elastic instability in heterogeneous HEA lattices:

"The structural response of crystalline solids to external stimuli such as heat and load generally determines their physical and mechanical behaviors including melting, incipient plasticity and fracture¹⁻⁴. High-entropy alloys (HEAs) contain multiple principal constituents and possess a scattering-distributed bond length due to their large atomic size mismatch and chemical complexities^{5,6}. Due to the severe heterogeneity of chemical bonds, HEAs are

considered to have intrinsically different structural responses to the external stimuli as compared with traditional dilute alloys. However, studies have shown that the incipient plastic deformation mechanism of bulk HEAs is still mediated by dislocation-related behaviors^{7,8}, which is in alignment with dilute alloys. Those unexpected results observed in bulk HEAs may be attributed to the fact that the existence of unavoidable crystallographic defects, such as dislocations and vacancies, conceals inherent elastic instability of HEAs.

Reducing the sample size to nanoscale or even smaller could be an effective way to minimize pre-existing defects and decipher the intrinsic lattice response to external stimuli^{2,9-11}. Specifically, in this work, the lattice evolution of nanosized HEA samples was investigated as a function of strain using *in situ* transmission electron microscopy (TEM). A typical method, *in situ* welding^{12,13}, was used for synthesizing nanoscale HEA samples with a dog-bone shape. The nearly defect-free HEAs fabricated are desirable samples for *in situ* atomic resolution characterization and mechanical testing. Using these nanoscale HEAs, we observed a distinct manifestation of elastic instability termed "elastic strain-induced amorphization", thereby broadening the conventional understanding beyond the classic framework of dislocation-mediated incipient plasticity. We further found that the local lattice periodicity vanishes as the elastic instability occurs at an elastic strain of ~10%, which intrinsically differs from the amorphization process induced by defect accumulation. The latter has been reported in ordered TiNi alloys, covalent materials and most recently bulk HEAs¹⁸⁻²⁴. Our findings offer new insights into the understanding of elastic instability and atomistic mechanisms for basic physical and mechanical phenomena such as elastic instability and incipient plasticity."

The reviewer's explanation of defect activity in nanoscale samples is entirely correct and consistent with our discussion in the manuscript. Our deliberate choice of using small-scale samples is grounded in their intrinsic difficulties on defect premature nucleation, a necessary condition for triggering the novel lattice instability behavior (i.e., elastic strain-induced amorphization). Notably, it is only in nanoscale multi-component alloys that we observed strain-induced amorphization, whereas traditional dislocation-related behaviors predominantly govern lattice instability in nanoscale binary and elemental metals. Therefore, in the nanoscale regime, we assert the discovery of a novel elastic instability in nanoscale multi-component

alloys, characterized by a solid-state amorphization process devoid of dislocation involvement.

We hope that our thorough substantiation of "melting" and explanation on size effect will address and dispel any misunderstandings held by the reviewer. This unusual solid-state amorphization observed in nanoscale multi-component alloys, characterized by large elastic strain rather than defect accumulation, can be regarded as a new mode of elastic instability, which enhances our profound understanding of fundamental aspects related to incipient plasticity.

Reviewer #4 (Remarks to the Author):

Overall, this reviewer also agrees with the opinion of Reviewer #3 that the observed phenomenon is closer to solid-state amorphization than melting. The basis on which the authors claim the phenomenon as melting are: i) the mechanical strain reaches 10%, which is comparable to the Lindemann criterion for melting, ii) there were no defects accumulation that mediates the solid-state amorphization, and iii) occurrence of plasticity was suppressed due to the inhomogeneous nature of atomic structures in HEAs. However, none of these three pieces of evidence look sufficient to fully support the authors' claim for strain-induced melting of the HEAs. Here are the reasons:

1) As the authors mentioned in the introduction, the Lindemann criterion states that melting would occur when the root mean square of the atomic "thermal" displacement exceeds ~ 10%. Given that the criterion is formulated in terms of root mean squared displacement, not by the average displacement, it usually assumes the stochastic atomic motion, giving zero net atomic displacements. However, the mechanical atomic displacements in this work are all aligned along the tensile direction, so it is unclear how the two distinct phenomena having different underlying assumptions can be combined.

Authors' response:

We appreciate the reviewer's pertinent comments and acknowledge that referring to solid-state amorphization as "melting" is inappropriate. To address this concern, we have replaced the term "strain-induced melting" with "elastic strain-induced amorphization" and subsequently reshaped the Introduction of our manuscript. The emphasis has been redirected towards unveiling a new manifestation of elastic instability in heterogeneous HEA lattices. It is worth noting that the solid-state amorphization observed in nanoscale multi-component alloys was triggered instantaneously when the local elastic strain approached ~10%, surpassing the conventional understanding of solid-state amorphization through defect accumulation. This elastic strain-induced amorphization represents a novel mode of elastic instability, extending the traditional understanding beyond the classic framework of dislocation-mediated incipient plasticity.

Following is the reshaped introduction:

"The structural response of crystalline solids to external stimuli such as heat and load

generally determines their physical and mechanical behaviors including melting, incipient plasticity and fracture¹⁻⁴. High-entropy alloys (HEAs) contain multiple principal constituents and possess a scattering-distributed bond length due to their large atomic size mismatch and chemical complexities^{5,6}. Due to the severe heterogeneity of chemical bonds, HEAs are considered to have intrinsically different structural responses to the external stimuli as compared with traditional dilute alloys. However, studies have shown that the incipient plastic deformation mechanism of bulk HEAs is still mediated by dislocation-related behaviors^{7,8}, which is in alignment with dilute alloys. Those unexpected results observed in bulk HEAs may be attributed to the fact that the existence of unavoidable crystallographic defects, such as dislocations and vacancies, conceals inherent elastic instability of HEAs.

Reducing the sample size to nanoscale or even smaller could be an effective way to minimize pre-existing defects and decipher the intrinsic lattice response to external stimuli^{2,9-11}. Specifically, in this work, the lattice evolution of nanosized HEA samples was investigated as a function of strain using *in situ* transmission electron microscopy (TEM). A typical method, *in situ* welding^{12,13}, was used for synthesizing nanoscale HEA samples with a dog-bone shape. The nearly defect-free HEAs fabricated are desirable samples for *in situ* atomic resolution characterization and mechanical testing. Using these nanoscale HEAs, we observed a distinct manifestation of elastic instability termed "elastic strain-induced amorphization", thereby broadening the conventional understanding beyond the classic framework of dislocation-mediated incipient plasticity. We further found that the local lattice periodicity vanishes as the elastic instability occurs at an elastic strain of ~10%, which intrinsically differs from the amorphization process induced by defect accumulation. The latter has been reported in ordered TiNi alloys, covalent materials and most recently bulk HEAs¹⁸⁻²⁴. Our findings offer new insights into the understanding of elastic instability and atomistic mechanisms for basic physical and mechanical phenomena such as elastic instability and incipient plasticity."

2) *Even though the authors claim there are no defects, e.g., dislocation, that can mediate the solid-state amorphization, the samples actually have them, i.e., the free surface. Given the very small sample size in this work, the authors may need to verify that the role of the free surface is negligible to mediate amorphization.*

Authors' response:

We appreciate the reviewer's thoughtful consideration. As the reviewer correctly pointed out, the substantial surface stress resulted from the high surface-volume ratio in nanoscale samples significantly influences the mechanical behavior. Examination of Figs. 1c1 and c2 reveals that amorphization in nanoscale HEA initiates from the free surface and gradually propagates into the nanoscale sample. Therefore, the free surface acts as a nucleation site for such elastic strain-induced amorphization, driven by the high surface stress in such nanoscale specimens. In nanoscale binary and elemental metals, free surfaces also serve as nucleation sites for elastic instability, leading to the emission of dislocations from the free surfaces. However, it is worth noting that no elastic strain-induced amorphization, as observed in nanoscale HEA samples, was detected in these materials. Hence, the free surface serves as a nucleation site for the elastic instability of the nanoscale alloys, but its influence on the elastic instability pattern determined by the intrinsic characteristics of the lattice is limited. To comply with the reviewer's suggestions, nevertheless, we have added a demonstration on the role of the free surface in Line 11-13, Page 4:

"With further tension, the periodic crystalline lattice was disrupted and a transition from BCT to amorphous structure occurred, as shown in the red square region of Fig. 1c1. Note that the free surface serves as the nucleation site for such elastic instability."

3) In the supplementary discussion, the authors claim that the electron beam irradiation is negligible because i) no knock-on damages were observed, and ii) the temperature rise is negligibly small. However, there is an additional effect the authors may need to consider, i.e., under electron beam irradiation, the atoms in the metallic materials get continuously ionized, which would alter the properties of the HEAs far from their ground-state values. The easiest way to verify the electron irradiation effects are negligible is by performing additional tensile experiments in the TEM with the electron beam off during tensile deformation. If the authors observe the same kinds of amorphization (or melting according to the authors' claim) with beam-off after ~ 10% tensile elongation, they can safely claim that there were no electron beam effects.

Authors' response:

We appreciate the reviewer's kind concern. As suggested, the tensile test under the beam blank condition is indeed crucial for mitigating the effects of electron beam irradiation. In our original manuscript, we have carefully addressed this concern by conducting tensile testing on nanoscale HEA with the electron beam switched off. One typical example, illustrated in Fig. S4 and discussed in Supporting discussion 1, revealed that amorphization still occurred under these conditions. Therefore, we firmly conclude that the influence of electron irradiation on the initiation of amorphization can be ignored.

4) The authors' analysis of the dislocations is confusing. First, using the DFT calculations, they proved that the atomic structures in the HEAs are inhomogeneous (though this claim itself is somewhat obvious). However, they did give a reasonable explanation of how this structural inhomogeneity contributes to suppressing the dislocation plasticity. The authors' analysis based on the Frank-Read model does not seem to be appropriate here. Generally, in the nano-sized sample, there is not enough space to accommodate two pinning points to create the Frank-Read (FR) source, so usually, the FR source does not exist. Anyhow, the authors also claim there was no dislocation in their samples. The FR source is basically a dislocation segment with two pinning points. The authors would be better off focusing on the surface nucleation of dislocations and need to convincingly convey how the atomic-structural heterogeneity induces the suppression of dislocation nucleation.

Authors' response:

We appreciate the reviewer for the constructive comments and acknowledge that using the Frank-Read source model to describe dislocation nucleation in nanoscale samples is inappropriate. Instead, we have adopted the "hot spots" model, commonly employed to elucidate dislocation nucleation in nanoscale samples without Frank-Read sources. This model helps to comprehend the challenges associated with dislocation nucleation in nanoscale HEA samples characterized by chemical short-range ordering and extensive atomic-scale heterogeneities (Line 21-22 in Page 12, Line 1-21 in Page 13 and Line 1-6 in Page 14).

" The difficulty in dislocation nucleation in nanoscale HEAs can be qualitatively accessed by the "hot spots" model, where atoms with relative displacements above a critical value (referred to as 'hot spots') are considered precursors to lattice defect nucleation. The critical elastic shear strain (γ)

derived from this model for dislocation nucleation can be expressed as follows:

(1)

where, $A(T)$ and $B(T)$ are functions of temperature (T) only. $A(T)$ represents the atomic displacements along the straining direction while $B(T)$ accounts for the isotropic and random atomic displacements due to thermal agitation. b is the magnitude of the Burgers vector.

γ_c represents the additional thermal agitation component and is a function of the strain rate ($\dot{\epsilon}$) and temperature. α and m are constants related to the critical displacement value and the probability distribution of atomic displacements in the crystal, respectively. N is the number of atoms in the efficient volume for defect nucleation, which directly links the critical elastic shear strain of dislocation nucleation to the sample size and packing factor of lattice. Namely, if the sample size or packing factor is increased, the first term in curved brackets in Eq. (1) would decrease, and the critical strain should decrease, and vice versa.

In nanoscale HEAs characterized by chemical short-range ordering due to extensive atomic-scale heterogeneities, the volume of dislocation nucleation is further subdivided, as schematically shown in Fig. 4e (where D and d represent the diameter of the sample and the efficient volume for dislocation nucleation, respectively). Consequently, N is dramatically decreased (even compared with that in nanoscale dilute alloys), and thus the critical elastic shear strain of dislocation nucleation (γ_c) is extremely high in the nanoscale HEAs. On the other hand, due to the non-close-packed lattice structure, the packing factor of BCC lattices is lower than that of FCC lattices, indicating that BCC lattices usually possess fewer atoms per unit volume and correspondingly higher resistance to dislocation nucleation. As a result, even at the high elastic strain, dislocation nucleation does not take place in nanoscale BCC HEAs, and eventually the elastic strain-induced amorphization occurred."

Overall, the evidence on which the authors claimed strain-induced melting does not seem to be convincing enough. Instead, there is a risk of misleading the readers of this article.

Authors' response:

We appreciate the thoughtful considerations raised by the reviewer. In response to the misunderstanding related to our nomenclature, we replaced the term "strain-induced melting" with "elastic strain-induced amorphization" and accordingly reshaped the Introduction part of our manuscript. We believe that this change clarifies the phenomenon under discussion while preserving the core innovation of the manuscript: the discovery of a new mode of elastic instability through elastic strain-induced amorphization, which is fundamentally different from defect accumulation-mediated amorphization, expands traditional understanding beyond the typical framework of dislocation-mediated incipient plasticity. We hope that our careful revisions can address the concerns raised by the reviewer.

REVIEWERS' COMMENTS

Reviewer #2 (Remarks to the Author):

After changing the term 'melting' to 'amorphization' and verifying the accuracy of its use, the paper is ready for publication.

Reviewer #3 (Remarks to the Author):

The authors have carefully addressed the reviewers' concerns, in particular regarding the melting and solid state amorphization. The toning of the observed phenomena toward solid state amorphization really leads the work to a proper standing for the community. It is apparent that, one of the pre-conditions for the present observation is the reduced size due to the requirement of the TEM sample, which, for general readers, could be regarded as an artifact, but essentially rather an unintentional advantage for making this observation possible. To make this point crystal clear, I would suggest the authors modify the following paragraph in the manuscript to clearly point out the fact the present observation is associated with the small scale of TEM sample, an unintentional advantage.

"The discovery of the elastic strain-induced amorphization phenomenon is attributed to the extremely large elastic strain ($\sim 10\%$) in the nanosized HEA samples. However, the elastic limit in bulk samples is usually less than 2% due to the ease of dislocation gliding. Therefore, although traditional defect accumulation-induced amorphization may occur in severely plastically deformed bulk samples—as evidenced by features such as amorphous bands in a 90% rolled bulk TiHfZrNb sample (as illustrated in Supporting Discussion 2 and Figs. S6 and 7), the elastic strain-induced amorphization cannot be observed in bulk sample due to premature dislocation generation."

Further, as the author has already done the large-scale deformation, the data can be added into the supplementary information. With above suggested clarifications, I get no objection for publication of the work.

Reviewer #4 (Remarks to the Author):

The authors reasonably addressed the concerns this reviewer raised previously, and thereby recommend the publication of this manuscript in Nature Comm.